# Seed Germination of Invasive *Phytolacca americana* and Potentially Invasive *P. acinosa*

**DOI:** 10.3390/plants12051052

**Published:** 2023-02-26

**Authors:** Simona Strgulc Krajšek, Aleš Kladnik, Sara Skočir, Martina Bačič

**Affiliations:** Biotechnical Faculty, Department of Biology, University of Ljubljana, Večna pot 111, SI-1000 Ljubljana, Slovenia

**Keywords:** *Phytolacca americana*, *Phytolacca acinosa*, pokeweed, invasive plant species, seed viability, fruit ripening, germination, plant reproduction

## Abstract

*Phytolacca americana* and *P. acinosa* are alien plant species in Europe. The former is considered invasive and more widespread. In order to develop effective and safe eradication and plant disposal methods, the present research focused on the seed germination of the two species. Fruits of different ripeness of both species were collected (fresh and dry seeds within and without pericarp), after which both the germination and maturation were tested. We also tested the continued maturing of fruits on cut plants and observed the development of fruits on whole plants with a cut taproot (in addition to when only the upper part of the stem with fruit racemes was cut off). In general, the seeds germinated from all stages of fruit ripeness, although the germination of dry seeds was better compared to fresh seeds. *P. americana*’s seeds germinated better and the fruit ripening on cut plants was also more successful compared to *P. acinosa*. These results could partly explain the invasive success of *P. americana*. According to our results, removing all fruiting plants from the eradication site is crucial regardless of the fruit development stage.

## 1. Introduction

Invasive alien plant species (IAPS) significantly impact many ecological variables on various levels: ecosystem, community, and species [1]. Thus, it is understandable that IAPS have been the subject of numerous studies: the number of published papers dealing with IAPS increased exponentially between 1980 and 2010 [2]. Information about the ecology, reproduction, development, and interactions of IAPS with other organisms is critical to understand the mechanisms behind their invasive success. Research on the reproduction strategies of IAPS is also essential to plan effective strategies to prevent or limit their spread and apply the most effective and safe eradication methods.

The present study focused on the seed maturation and germination of two pokeweed species introduced to Europe: *Phytolacca americana* L., which is already considered an invasive species; and *P. acinosa* Roxb., which is naturalised but has the potential for further spread. From the genus *Phytolacca*, these two species are present in Slovenia, where our research took place [3,4], while another three non-native species were recorded in other parts of Europe: *P. dioica* L. is naturalised in some Mediterranean countries, *P. polyandra* Batalin in Great Britain and NW France, and *P. heterotepala* H. Walter in Portugal. However, the most widespread species of the genus in Europe are *P. americana* and *P. acinosa* [5].

*P. americana* is a herbaceous perennial plant with a taproot. The stem is branched, up to 3 m high, hollow inside, and usually purple-coloured. Leaves are alternate, petiolate, oblong- to ovate-lanceolate and up to 30 cm long. Flowers are pentamerous and hermaphroditic, have 10 stamens and 10 carpels, and grow in simple 5–15 cm long racemes. The fruit is a 10-seeded berry that is dark purple to black. The fruit racemes are pendulous. The seed is smooth (not furrowed) and shiny black, reniform-orbicular in shape, and measures 2.5–3.0 mm in diameter [6,7]. The species originated from North America. It was brought into Europe in the 17th century and cultivated as a dye plant [6]. It spread rapidly in central Europe and mainly inhabits the open areas in forests as well as in cuttings, below power lines, and in forest clearings and ruderal sites [8]. *P. americana* has been considered to be regulated by the EU IAS legislation [9,10] but was not included in the list of invasive alien species of Union concern [11].

*P. acinosa* is morphologically very similar to *P. americana*. Stems of *P. acinosa* are usually lower and not or only sparsely reddish. There are only eight stamens and eight carpels in flower. Fruits are eight-seeded, and carpels are free also when the fruit is ripe, so the fruit is deeply furrowed. The racemes of *P. acinosa* are mostly erect during flowering and fruiting [6,7]. The species originated from East Asia. Recently, it has been reported in many parts of Europe as a casual species that mainly spreads in urban and suburban areas [6].

*P. americana* is easy to distinguish from *P. acinosa*, especially on basis of their fruits. Their differences also include flowering time: *P. acinosa* starts flowering from May onwards, whereas in Western Europe *P. americana* does not flower before July [12].

Pokeweeds (*Phytolacca* spp.) reproduce sexually, and their dark-purple to almost black fruits attract frugivorous birds that disperse the seeds [13]. Very little is known about the germination of *P. acinosa*, while the seed germination of *P. americana* is already well researched. For example, the seeds of *Phytolacca americana* remained viable for decades [14], but a reduction in germination with the age of seeds was observed [15]. Several experiments investigated the dormancy of *P. americana* seeds. It was shown [16] that stratified and dry-stored seeds germinated in light. In the dark, germination was less successful for stratified seeds, and non-stratified dry-stored seeds did not germinate. The seed testa was thick, and it was suggested that unstratified seeds germinate if the seed coat is mechanically ruptured [17]. One of the methods of scarification of the seed coat is soaking in acid, a procedure that resembles the passage of the seed through the digestive tract of birds. However, it was found that there was no significant difference between the germination of seeds collected directly from plants and those collected from bird droppings found in the vicinity of the sampling site [18]. However, in laboratory conditions the percentage of germinated seeds was higher when seeds were soaked in 95% H_2_SO_4_ for more than 5 min [15,16].

For *P. americana*, the percentage of germinated seeds varied widely among published experiments [15,16,18]. Even within a single survey [18], considerable discrepancies were observed in which the variation in germination ranged from 25 to 99%. It was assumed that the variability could depend on the ripeness of the fruits, although the germination was not statistically related to seed weight. Some authors suggested that this widely distributed species may have latitudinal differences in seed germination [15]. The variability in the germination of seeds of *P. americana* collected from different individuals was also observed [16]. Another reason might be the non-comparable germination protocols from different publications [15,16,18]. The critical factor is temperature because seeds germinated better at temperatures over 24°C [16]. Some authors obtained better germination results when seeds were treated with gibberellic acid and germinated during an alternating light–dark regime [17]. However, in another study gibberellic acid only had a positive effect in combination with 95% H_2_SO_4_ treatment and dark conditions [16]. Better germinating results were obtained at neutral pH compared to acidic conditions (pH 3 or 5), and seeds did not germinate at pH 9 and 11 [17]. Further observations showed that the extracts of immature and ripe berries, pigmented stems, and mature leaves of *P. americana* reduced the germination of its seeds [15], and it was suggested that the autotoxicity of berry juice prevents the autumnal germination of *Phytolacca* seeds.

In contrast to the many published *P. americana* germination studies, there are almost no data about *P. acinosa* seed germination. The only accessible published experiment was from China [19], where the best germinating results were obtained when seeds were soaked in 98% H_2_SO_4_ for 10 min and 500 mg/L gibberellins for 1 h, heated in a water bath at 40 °C for 2 h, and inoculated in a solid medium. The seeds germinated in a dark box at 25 °C at 75% relative humidity, and the average 7-day germination rate was 90%.

Understanding the biology of alien species is essential to prevent or predict their invasion. Hence, information about the reproduction strategies of IAPS is essential to prepare effective and safe protocols for eradicating plants from nature outside their native distribution ranges, while it also helps to avoid unnecessary costs in land management. Considering *Phytolacca* species as IAPS, it is important to know at what stage of fruit ripeness seeds become able to germinate and how successful fruit ripening and seed maturation is on cut plants. As *P. acinosa* is a neglected alien species but is gradually spreading to natural habitats in Europe, it is important to research these aspects and compare its biology to the more researched *P. americana*.

The aims of our study were: (i) to analyse seed germination of fresh and dry seeds from different stages of fruit ripeness of *P. americana* and *P. acinosa* and test if stratification influenced the germination success, (ii) to analyse the germination of seeds not isolated from fruits, (iii) to monitor the fruit ripening on cut *Phytolacca* plants and test the germination of seeds from these fruits, and (iv) to analyse the difference in habitat preferences and invasiveness between *P. americana* and *P. acinosa* in Slovenia.

Additionally, the results were used for preparing protocols for plant eradication and disposal of collected plant material to prevent the unintended further spread of these alien species and to consider whether the differences in the germination of the two species could help us explain the greater invasive success of *P. americana* in Slovenia.

## 2. Results

### 2.1. Germination of Seeds of Different Ripeness

#### 2.1.1. Germination of Fresh Seeds

In experiment 1, we tested the germination of intact fresh seeds immediately after extraction from fresh fruits of different ripeness. The results of the germination of *Phytolacca americana* (PAM) are shown in Figure 1A. In 42 days, of 150 sample seeds extracted from black fruits, 47% germinated and 38% developed unfolded cotyledons. This germination result was statistically higher than the germination of seeds extracted from immature red and green fruits. Within 42 days, 20% of seeds from red fruits germinated and 11% unfolded cotyledons. In the same period, 18% of seeds from green fruits germinated and 13% unfolded cotyledons.

Only one fresh seed of *P. acinosa* (PAC) germinated in this experiment (Figure 1B). It was extracted from an immature, red-coloured fruit; germinated on day 7; and the cotyledons unfolded on day 12. Due to the poor germination result of *P. acinosa*, we randomly chose 60 ungerminated seeds from every series (black fruits, red fruits, and green fruits) and conducted a tetrazolium viability test. The percentages of potentially viable seeds were 93.3%, 63.3%, and 48.3%, for PAC-B, PAC-R, and PAC-G seeds, respectively.

Intact fresh seeds of *P. americana* (Figure 1A) in all phases of fruit ripeness germinated better than seeds of *P. acinosa* (Figure 1B).

In experiment 2, we treated seeds with 95% sulfuric acid before the germination test. Only a few seeds of *P. americana* germinated within 39 days (Figure 1C), and the percentage of germinated seeds was lower compared to untreated intact seeds (Figure 1A). None of the 95% sulfuric acid-treated seedlings unfolded cotyledons.

The treatment with 95% sulfuric acid in the case of *P. acinosa* (Figure 1D) resulted in 7.7% of germinated seeds isolated from green fruits, 4.5% from red fruits, and 2.0% from black fruits (Figure 1B). Four seedlings from green and red fruits and none from black fruits unfolded cotyledons.

#### 2.1.2. Germination of Dry Seeds

In experiment 3, we tested the germination of dry seeds stored in a dry place at room temperature for about 6 months. The results of the germination of *P. americana* (PAM) are shown in Figure 2A. Within 42 days, 51% of the seeds extracted from ripe black fruits germinated and 33% developed unfolded cotyledons. The percentage of germinated seeds from red and green fruits was similar: 31% of seeds from red fruits germinated and 19% unfolded cotyledons, while 29% of seeds from green fruits germinated and 14% unfolded cotyledons.

The seeds of *P. acinosa* extracted from black fruits germinated in 46% (Figure 2B), and all seedlings developed unfolded cotyledons. The germination percentage of seeds from immature fruits was low: 4% isolated from red and 2% from green fruits.

In experiment 4, we tested the germination of dry seeds kept in a fridge at 6 °C for approximately 6 months. The germination results are shown in Figure 2C for *P. americana* (PAM) and in Figure 2D for *P. acinosa*. There were no statistically significant differences between the germination of seeds from red and black fruits. Still, the seeds from green fruits kept in the fridge germinated statistically better than those kept at room temperature (Figure 2A) or fresh seeds (Figure 1A). In addition, the seeds of *P. acinosa* isolated from black fruits stored at room temperature had the best germination result (46%; Figure 2B). The seeds from immature fruits germinated poorly after both storage methods. Almost all seedlings also unfolded the cotyledons until the end of the experiment.

#### 2.1.3. Tetrazolium Test for the Viability of Seeds

The seeds that did not germinate in experiments 1 (PAC only), 3, and 4 were further subjected to a tetrazolium viability test. In Table 1, it is noticeable that the percentages of potentially viable seeds were much higher than those of germinated seeds.

### 2.2. Germination of Seeds Not Isolated from Fruits

The results of the germination of seeds that were not previously isolated from fruits are shown in Figure 3. Whole fruits were planted in germinating trays filled with soil. In the first series (experiment 5, F), fresh fruits were sown immediately after the collection in August (solid columns in Figure 3), and in the second series (experiment 6, D) dry fruits were sown 6 months later (dotted columns in Figure 3). In the case of fresh fruits, only a few *P. americana* seeds germinated. In the case of dry fruits, germination was observed for both species, but more seeds of *P. acinosa* germinated (Figure 1B). However, the germination success of these two experiments was much lower than the germination of fresh or dry seeds previously isolated from fruits (Figure 1 and Figure 2).

### 2.3. Fruit Ripening on Cut Plants

The fruit ripening was observed on selected racemes on the whole plants and the upper part of the shoots of *Phytolacca americana* and *P. acinosa*. The fruits ripened until day 10 after collection, then the fresh pericarps began to dry, and no additional colour changes in the fruits were observed. As an example, a series of photographs of the selected raceme of *P. acinosa* is shown in Figure 4A. The change in colour of fruits was visible on day 3 and day 6 but a little less obvious on day 10, while the raceme remained unchanged in the following observation period.

The ripening of fruits in the racemes of both species is summarised in Figure 4B. In most cases, some fruits matured and changed their colour from green to red or red to black. Thus, ripening did not stop when the plants were cut. A difference in fruit ripening was observed between the whole plants with cut tap roots and the plants on which only the upper parts of the shoots were cut. More racemes on the whole plants developed with at least some completely mature, black-coloured fruits. In four cases of *P. acinosa* in which all fruits in the racemes were green on the first day of the observation, the fruits did not mature and remained green until the end of the experiment. The same situation was observed on one raceme of *P. americana* only.

### 2.4. Species Distribution and Their Spread in Slovenia

*P. americana* is more widespread outside gardens in Slovenia compared to *P. acinosa* (Figure 5). Before the year 2000, *P. americana* was recorded for only 16 Central European MTB quadrants [20] and was scattered in various parts of the territory, with the number of records per quadrant ranging from one to eight. In the following decade, the species spread from the first known localities and newly established places: it was documented to thrive in over 20 new quadrants with one to three records per quadrant. Since 2010, the species can be found in all parts of Slovenia. It was recorded in 2/3 of the quadrants covering the whole territory of Slovenia, with 1 to 159 records per quadrant. Hence, the species is strongly present in Slovenia, and only the alpine region seems less invaded.

The first observation of *Phytolacca acinosa* in Slovenia was published in 1998 for Ljubljana (Žale Cemetery), where the population was designated as “cultivated but not established” [21]. Starting from only one known locality of *P. acinosa* before 2000, the considerable leap in its observation is shown in Figure 5B. The number of localities per quadrant ranged from 1 to 50. Ljubljana had the highest concentration of the records (9953/1, 2, 3), with over a hundred recorded until 2022. In the majority of quadrants in other parts of the territory, there were only one or two known localities of the species.

## 3. Discussion

*Phytolacca americana* and *P. acinosa* are alien species in Europe that reproduce using seeds and are spread by both bird and human activities [6,8]. To understand the mechanisms behind the invasive success of IAPS, it is essential to have sufficient information about their biology, especially about their reproduction strategies. This information is needed to plan effective strategies to prevent the spread of IAPS and to apply the most effective, safe, and cost-beneficial eradication methods. Ineffective or unavailable eradication methods have been listed as a problem for several IAPS [8,22].

In plants that primarily reproduce sexually and spread through seeds, the following traits can enhance their invasive potential: the seeds are abundant and germinate well (and in some cases, even if the fruits are not entirely ripe, the fruits (and seeds) can continue ripening even if a plant is damaged), and the seeds are dispersed effectively at a great distance and germinate at an appropriate time so that the new plants have better chances of survival [23,24,25]. Our germination results elucidated some of the above-mentioned issues and enabled us to compare the two pokeweed species with the knowledge that *P. americana* is more invasive.

All the seeds in the present study that were extracted from ripe (black) or almost ripe (red) fruits of *P. americana* and *P. acinosa* were the same size with a comparably thick black testa. The green fruits contained seeds of different ripeness and ranged from tiny soft seeds with almost transparent testa to seeds that looked completely developed and were superficially similar to those in mature fruits. We assumed that the *Phytolacca* seeds’ development in both species was completed before the fruits were ripe, and the results of both the germination and the seed viability tests confirmed this prediction. Seeds isolated from green fruits of both species germinated in at least some of the experiments, and the viability tests showed that 40 to 100% of tested seeds were viable (Table 1).

Both species’ seeds germinated well when extracted from the pericarp. In natural conditions, the pericarp is removed when birds eat the fruits, when the soft pericarps are digested [13], or during the natural decomposition when fruits drop off the plant. For *P. americana*, it was reported that the pericarp contained substances that inhibited the germination of the seeds [15]. Our results were consistent with this report: we also observed that the germination of seeds not extracted from the fruits was much lower than that of extracted seeds. However, some seeds of *P. americana* did germinate after they were sown with intact fruits of different ripeness immediately after their collection in August. It was also observed for both species that sown dry berries of different ripeness germinated.

The fruits of both *Phytolacca* species continued to develop on cut plants even when the upper part of the stem with inflorescences was cut. Similar results have been observed for other species; for example, *Lupinus polyphyllus* [26] and *Alliaria petiolata* [27], from which fruits matured on a cut or pulled plants and seeds germinated. Thus, seeds were able to use stored resources for their development.

In addition to the germination experiments, our study also focused on the invasiveness of the two species in Slovenia—their current distribution and spread in the last decades. The results revealed the need to limit the rapid spread of both species as soon as possible.

*P. americana* is a highly competitive plant due to its high seed production, photosynthetic capacity, morphological and physiological adaptability, and effective use of resources [28]. The rapid increase in *P. americana* records in Slovenia in the last two decades (Figure 5A) reflects the rapid spreading of the species, but could be partly attributed to the improved gathering of distribution data from a larger community of observers [4]. The results of our germination experiments explained the invasive potential of the species: the seeds collected from Slovenian natural settings showed enough germination potential to enhance the establishment of new populations. According to our data, the species thrived on similar sites to those of *P. acinosa* but showed an even more extensive ecological range; it was documented to grow along roads, on forest and meadow edges, in vegetable fields, and even in dense invasive *Fallopia* stands. The observers reported that the plants developed dense stands in some places, while in other localities the population consisted of only a few plants. In Slovenia, it was observed that the species is less frequent in natural forests and occurs mainly in open areas (very frequently under power lines) [8]. Similar results were obtained in research from China, where it was found that *P. americana* was less successful in natural forests [28].

When discussing the distribution of *P. acinosa* and its increased number of observations in the last decades (Figure 5B), it must be noted that nowadays, the species is also better recognised by field botanists (in the past, it was often confused with *P. americana*). The awareness of its presence in Slovenia as well as the efforts in gathering distribution data of IAPS in Slovenia in general have increased [4]. According to our observations, the data from herbarium labels, and the observations by other sources, the species seems to thrive in (mostly) urban forests (disturbed sites, clearings, gaps left by a windstorm, etc.), urban wastelands (it can even grow from cracks in the walls), gardens, parks (under trees or shrubs), hedges, playgrounds, rest areas by the motorway, and cemeteries. The observers frequently reported that young plants developed from seed were found around the cultivated plants. They might have been sown by birds or—as shown by our germination experiments—developed from the seeds of dropped fruits. The populations without cultivated plants in the vicinity were probably sown by birds, which was also observed for *P. americana* [13]. Due to the rapid accumulation of *P. acinosa* observations in the last years, the species must be considered fully naturalised and potentially invasive in Slovenia.

The results of our germination experiments coupled with previously known information on the species biology gave us a firm basis for developing the guidelines and warnings regarding the eradication and disposal of collected *Phytolacca americana* and *P. acinosa*. In addition to the effective removal of the plants (without the possibility of regrowth and without unnecessary work and costs), it is crucial to prevent the unwanted dispersal of plant diaspores during eradication.

*P. americana* is more widespread in Slovenia (Figure 5A), and several eradication actions have been organised in the last few years. Cutting the plants did not work efficiently, but early evidence indicated that pulling the saplings and digging up the plants where the tap root was cut ca. 10 cm below the ground surface were effective [8,29]. During the eradication of fruiting plants, it is considered a high risk that the ripe berries will fall off the plant, and if the end of the vegetation season is warm enough, seeds of *P. americana* could start to germinate. Further field studies are needed to determine whether seedlings emerge in autumn and survive until the following season.

We observed that the seedlings began to store nutrient substances in the tap root early in the development of the plant; therefore, in favourable seasons such seedlings may overwinter. As *P. acinosa* seeds from fresh fruits did not germinate, we assumed this scenario was possible for *P. americana* only. Based on that information, we suggest that fruit racemes should be cut off the plant, bagged, and discarded so the seeds do not reinfest the soil [29,30].

Since cutting off the fruit racemes from plants is time-consuming, it is vital to know at what stage of fruit ripeness the seeds are already mature enough to germinate to avoid unnecessary work and costs. Not only are the seeds from ripe fruits problematic, but the seeds from green fruits also can germinate. Thus, it is vital to cut all fruit racemes from plants during the eradication and discard them safely to prevent plant dispersal. However, to reduce the costs and the time needed for cutting fruit racemes, it is more efficient to eradicate *Phytolacca* before the plants begin to fruit.

To summarise our findings relevant to a broader audience, we prepared a concise eradication protocol for *Phytollaca* for landowners, authorities, managers of protected areas, and the general public who are dealing with invasive pokeweeds (Figure 6).

The proposed protocol does not recommend using chemical eradication methods even though some of the herbicides proved to be effective and resource-efficient tools for treating *Phytolacca americana* [31]. Unlike in the case of invasive *Fallopia*, the stands of *P. americana* in Slovenia are never monospecific, meaning that native species thrive among the pokeweed plants. In such cases, the use of herbicides is not appropriate [31] and could be environmentally damaging. In Slovenia, the sustainable use of phytopharmaceuticals is promoted. The guidelines and criteria for the use of herbicides have been issued by authorities to reduce risks to and impacts on human and animal health and the environment [32]. Only an authorised person with completed training may use such phytopharmaceuticals.

The invasion of *P. americana* affects local fauna, and consequently, the environmental conditions and biotic interactions could be altered [33]. The species is also present in the urban environment and crop fields [34], which negatively impacts infrastructure and crop production. Regarding the results of our experiments and the invasive potential of *P. acinosa*, we assume that such negative impacts could soon be observed for both species of pokeweeds.

We found both species of *Phytolacca* still planted as ornamentals in some gardens in Ljubljana. The plants are attractive, and the people do not see them as problematic [31]. In recent years, several activities have been organised in Slovenia to educate citizens about IAPS. Some publications for the general public were printed [35] or published online [4,36], and within some project activities (for example, project Applause), several workshops have been organised. Hence, it was observed that public opinion on invasive alien plants began to change. The dispersal of IAPS by humans could be limited only by education and the raising of awareness. However, accurate and correct information based on scientific studies is crucial for achieving good practical results.

## 4. Materials and Methods

### 4.1. Plant Material

The plant material was collected from 5 locations in Slovenia:
*Phytolacca acinosa*: Slovenia: Ljubljana, Ljubljana Castle hill, by unpaved path Vozna pot to the Castle hill (46°02′39.6″ N 14°30′54.1″ E), among the bushes, 320 m a.s.l., Leg. and det.: S. Strgulc Krajšek, A. Kladnik, S. Skočir, 19. 8. 2020.*Phytolacca acinosa*: Slovenia: Ljubljana, Prule, by the street Zvonarska ulica and around the store Mercator (46°02′36.9″ N 14°30′30.6″ E and 46°02′36.9″ N 14°30′26.6″ E), planted and wildly growing in the park, 300 m a.s.l., Leg. and det.: S. Strgulc Krajšek, A. Kladnik, S. Skočir, 19. 8. 2020.*Phytolacca acinosa*: Slovenia: Ljubljana, Vič, between street Oslavijska ulica and the railway, between two pedestrian underpasses under the track (46°02′52.6″ N 14°29′20.9″ E), among the bushes, 300 m a.s.l., Leg. and det.: S. Strgulc Krajšek, 20. 8. 2020.*Phytolacca americana*: Slovenia: Ljubljana, Ljubljansko barje, Vnanje Gorice, by the unpaved road, south foothills of the hill Grič (45°59′39.7″ N 14°25′32.6″ E), among the bushes, 300 m a.s.l., Leg. and det.: S. Strgulc Krajšek, A. Kladnik, and S. Skočir, 19. 8. 2020.*Phytolacca americana*: Slovenia: Ljubljana, Bežigrad, abandoned land by street Vodovodna cesta between streets Triglavska and Posavskega (46°04′21.7″ N 14°30′18.0″ E), overgrown land, 300 m a.s.l., Leg. and det.: S. Strgulc Krajšek, A. Kladnik, and S. Skočir, 19. 8. 2020.

In August 2020, one whole plant of each species was collected with part of the taproot cut approximately 5 cm below ground, which should be done during the eradication of *Phytolacca* from natural habitats. Additionally, a few upper parts of the stem with partly ripe fruit racemes and cut fruit racemes of each species with fruits in different phases of ripeness were collected. Fruit racemes were stored in plastic bags in the fridge until use.

### 4.2. Seed Germination Tests

The extracted seeds from three different ripening phases of fruits were separated into green (G), red (R), and black (B), and then maceration of the fruits was conducted using a stainless-steel mesh sieve. The seeds were washed under running tap water to remove fragments of the pericarp. The first part of the extracted seeds was used immediately, and the remaining seeds were stored for later experiments.

We divided the germination tests into several experiments:

(1) Germination of fresh seeds: immediately after the seed extraction, the seeds were surface-sterilised by immersion in an aqueous solution of sodium hypochlorite (16.5 g/L, Arekina, Šampionka Renče) for 15 min. Afterwards, the seeds were rinsed three times in distilled water for 5 min and placed in 9 cm Petri dishes on two layers of sterilised filter paper and watered with 4 mL of distilled water. Generally, 6 replicates with 25 seeds for each sample were performed (or fewer when not enough seeds were available).

(2) Germination of fresh seeds treated with H_2_SO_4_ [15]: immediately after the seed extraction, the seeds were soaked for 10 min in concentrated H_2_SO_4_, after which they were rinsed three times in distilled water for 5 min and subsequently placed in 9 cm Petri dishes on two layers of sterilised filter paper and watered with 4 mL of distilled water. Generally, 6 replicates with 25 seeds for each sample were performed (or fewer when not enough seeds were available).

The Petri dishes from experiments 1 and 2 were enclosed in transparent plastic bags and left in a growth chamber at 24 °C, 60% humidity, and a 12/12 h light/dark cycle. The seeds were examined every day, and the germination was monitored. A seed was considered germinated on the day of the emergence of its radicle. Additionally, the opening of cotyledons was observed as a marker of further seedling development. When needed, distilled water was used for watering. Mould or bacterial infection occurred in some Petri dishes during the experiment, and in these cases, uninfected seeds were transferred to freshly prepared Petri dishes and subsequently treated identically. Experiment 1 lasted for 42 days, and experiment 2 lasted for 39 days.

(3) Germination of dry seeds kept at room temperature: dry seeds were stored in a dry place at room temperature for about 6 months.

(4) Germination of dry seeds kept in a fridge: dry seeds were stored in a refrigerator at 6 °C for about 6 months.

Sterilisation of seeds and germination experiments 3 and 4 were carried out identically to experiment 1. The experiments lasted for 42 days.

Randomly selected seeds that did not germinate during experiments 1, 3, and 4 were selected for the tetrazolium test of the viability of seeds. The seeds were cut in half, placed in the holes of microplates, then covered with a few drops of 1% triphenyl tetrazolium chloride (TTK) and kept in the dark at room temperature for 12 h. The embryos that were coloured dark red after the incubation were counted as potentially viable.

(5) Germination of seeds in fresh fruits: fruits were cut from the racemes and sorted according to ripeness. A total of 48 fruits were selected for each series (*P. americana* and *P. acinosa* green, red, and black fruits) to be used immediately. For the red fruits of *P. americana*, only 36 fruits were used due to lack of material. Whole fruits were planted in plant germination trays with transparent covers using sowing substrate (Humovit, Cinkarna Celje) consisting of a mixture of acidic white and alkaline peat, high-quality quartz sand, and water-soluble fertiliser. Fruits were sown in trays ca. 1 cm deep in the substrate, watered with tap water and placed in a growth chamber at 24 °C, 60% humidity, and a 12/12 h light/dark cycle for 3 months. The seeds were examined every 3 days to monitor germination. A seed was considered germinated on the day of the emergence of the seedling from the substrate.

(6) Germination of seeds in dry fruits: fruits were cut from racemes and sorted according to ripeness. A total of 36 fruits were selected for each series (*P. americana* and *P. acinosa* green, red, and black fruits) and left on trays in a dry room at room temperature until completely dry. The germination experiment was the same as explained for experiment 5.

Data analysis of all experiments was conducted using survival analysis in GraphPad Prism 5.01, which automatically compared curves representing different data sets (treatments) using the log-rank (Mantel–Cox) test.

### 4.3. Fruit Ripening on Cut Plants

The whole plants of *P. americana* and *P. acinosa* and the cut upper parts of the stems with partly ripe fruit racemes of both species were spread on the floor of a glasshouse and covered with plastic foil. Five racemes with fruits of different ripeness were selected on each plant (only four on the cut upper part of *P. acinosa*), marked, and photographed. The fruits of different colours (green, red, black) were counted on the 1st, 3rd, 6th, and 10th day of the experiment, and all racemes were photographed. The racemes were monitored until the 34th day of the experiment.

### 4.4. Species Distribution in Slovenia

The data for the distribution maps were obtained from 4 sources: (i) the Bioportal database [37] maintained at the Centre for Cartography of Fauna and Flora (CKFF) research institution. The database includes publicly available and private data from several sources (digitalised distributional data from the literature, observations from Slovenian field botanists, projects etc.). In addition, public data (mainly publicly funded) were also included in the database, but these are not necessarily easily accessible to the public; (ii) the data from the LJU herbarium at the Department of Biology, Biotechnical Faculty, University of Ljubljana. The herbarium material of the genus *Phytolacca* was revised, the species identifications were verified, and the data from herbarium labels were obtained; (iii) data from the “Spletni portal invazivke” web platform established in the project LIFE ARTEMIS (LIFE15 GIE/SI/000770) [4]; and (iv) the data gathered during our fieldwork for the present study.

The distributional data were sorted according to the observation date in 3 categories: before 2000, between 2000 and 2010, and after 2010 (to 2022) to present the spreading dynamics of the two species. The grid lines of the maps corresponded to the base fields of the Central Europe flora mapping (MTB quadrants; area of about 5 × 6 km) [20].

The complete data on the species distribution is available from the authors.

## 5. Conclusions

The seeds of both analysed *Phytolacca* species isolated from different stages of fruit ripeness germinated. Dry seeds germinated better than fresh seeds. The seeds kept in the fridge did not germinate any better than those stored at room temperature, so we concluded that the seeds were not dormant.The seeds that were sown together with pericarp germinated, but their germination was much lower than that of isolated seeds. The only seeds of *P. americana* that germinated in the experiment were from the sown fresh fruits. Some seeds of both species germinated when we used dry fruits previously kept at room temperature.The fruits on the cut plants of both *Phytolacca* species continued to ripen. More effective ripening was observed on the whole plant than on the cut upper parts of the shoots.In Slovenia, *P. americana* is more widespread than *P. acinosa*, and a rapid increase was observed in records for both species during the last two decades. The species thrive on similar sites; however, *P. americana* showed a more extensive ecological range.

The present results could partly explain the greater invasive success of *P. americana*. Additionally, the prepared protocol for plant eradication and disposal of collected plant material can contribute to prevent the unintended further spread of these two alien species. According to our results, removing all fruiting plants from the eradication site is crucial regardless of the fruit development stage.

## Figures and Tables

**Figure 1 plants-12-01052-f001:**
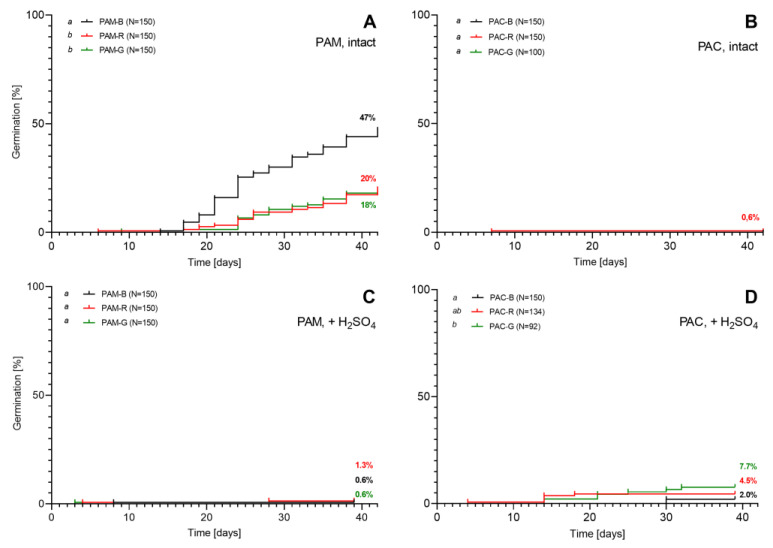
The germination of fresh seeds of *Phytolacca americana* (PAM; graphs (**A**,**C**)) and *P. acinosa* (PAC; graphs (**B**,**D**)) in Petri dishes immediately after the extraction from fresh fruits of different ripeness: immature (green (G) and red (R)) and ripe (black) (B). Graphs (**A**,**B**) show the germination of sterilised seeds without pre-treatment (experiment 1). Graphs (**C**,**D**) show the germination of fresh seeds after treatment with 95% sulfuric acid (experiment 2). A log-rank (Mantel–Cox) test was used for the comparison of the survival curves. Letters *a* and *b* in legends mark statistically different curves.

**Figure 2 plants-12-01052-f002:**
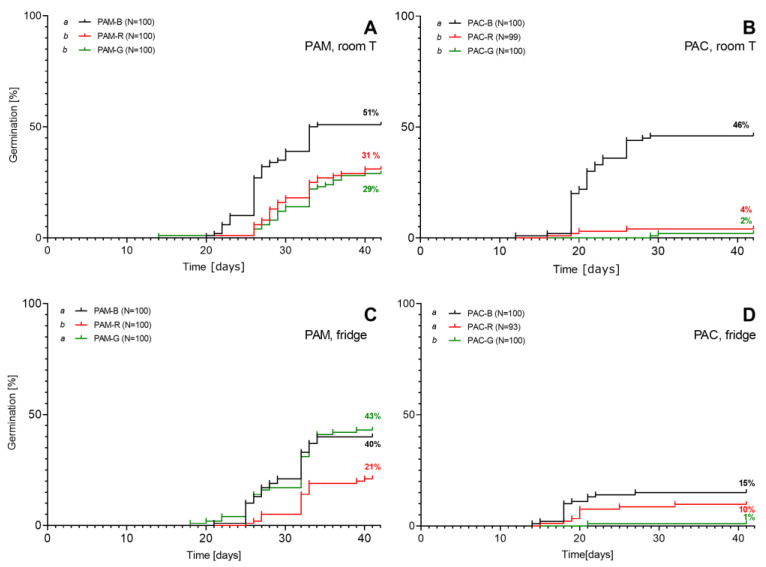
The germination of dried seeds of *Phytolacca americana* (PAM; graphs (**A**,**C**)) and *P. acinosa* (PAC; graphs (**B**,**D**)) in Petri dishes approximately 6 months after the extraction from fresh fruits of different ripeness: green (G), red (R), and black (B). Graphs (A) and (B) show the germination of sterilised seeds kept at room temperature (experiment 3). Graphs (**C**,**D**) show the germination of sterilised seeds kept in the fridge at 6°C (experiment 4). A log-rank (Mantel–Cox) test was used for the comparison of the survival curves. Letters *a* and *b* in legends mark statistically different curves.

**Figure 3 plants-12-01052-f003:**
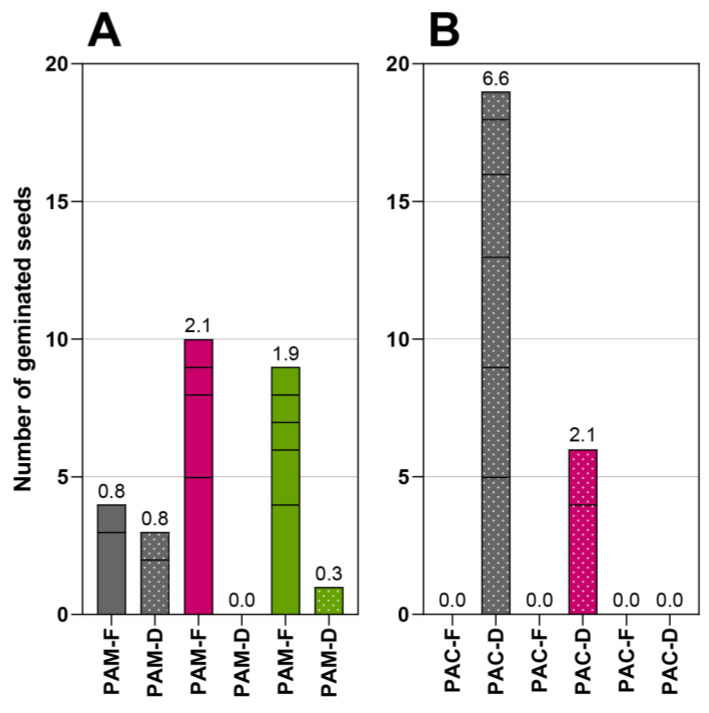
The number of germinated seeds from whole fruits of *P. americana* (**A**) and *P. acinosa* (**B**). The colours of the columns represent the colours of fruits. Solid columns mark fresh fruits (F) planted immediately after collection, and dotted columns mark dry fruits (D) planted 6 months after collection. The division lines within the columns indicate the number of germinated seeds that originated from the same fruit. The numbers above the columns show the percentage of germinated seeds.

**Figure 4 plants-12-01052-f004:**
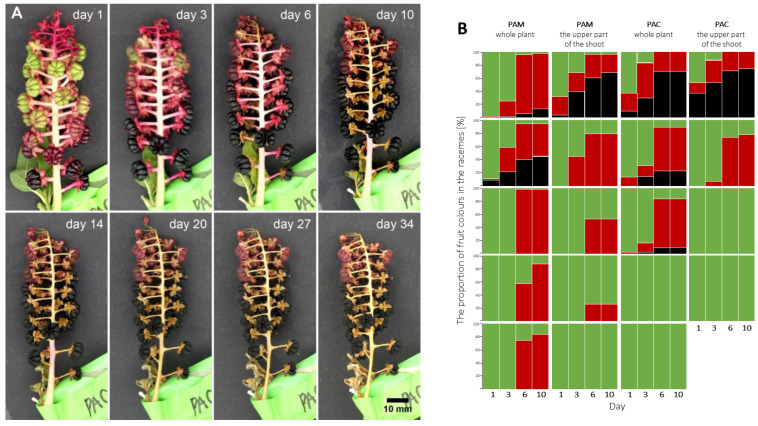
Fruit-ripening experiment on a whole dug-out plant with a cut tap root and a cut made to the upper part of the plant shoot. (**A**) Fruit ripening in the selected raceme of *Phytolacca acinosa*. (**B**) Summary of the fruit ripening of all observed fruit racemes. The group of 4 columns represents one fruit raceme in 4 days of observations, and the colours show the proportions of fruits of different ripeness (the colours of the columns represent colours of fruits). Legend: PAM—*Phytolacca americana*; PAC—*P. acinosa*.

**Figure 5 plants-12-01052-f005:**
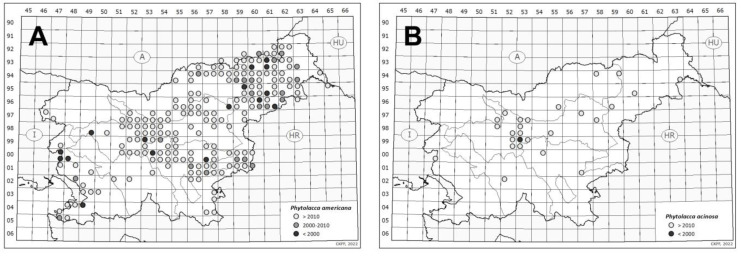
The MTB quadrants with data on *P. americana* (**A**) and *P. acinosa* (**B**) outside gardens in Slovenia. Black dots represent data up to 2000, grey dots from 2000 to 2010, and white dots from 2010 on.

**Figure 6 plants-12-01052-f006:**
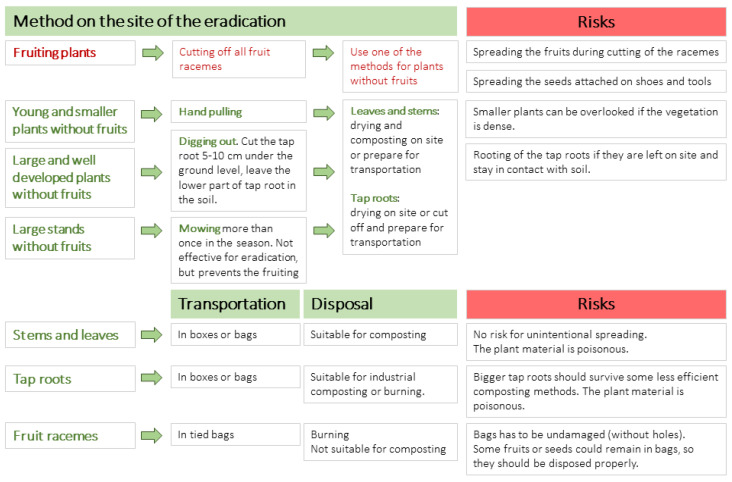
The protocol for eradication and disposal of collected *Phytolacca americana* and *P. acinosa* to prevent the unintended further spread.

**Table 1 plants-12-01052-t001:** The results of the tetrazolium test of ungerminated seeds from experiments 1, 3, and 4. The numbers represent the percentage of potentially viable seeds, and the numbers in brackets are the percentages of germinated seeds.

Fruit/Seed	Experiment 1	Experiment 3		Experiment 4	
Seeds isolated from	PAC, fresh (%)	PAM, room T (%)	PAC, room T (%)	PAM, fridge (%)	PAC, fridge (%)
Black fruits	93 (0)	80 (51)	90 (40)	50 (46)	55 (15)
Red fruits	63 (0.6)	95 (31)	100 (21)	75 (4)	75 (10)
Green fruits	48 (0)	100 (29)	90 (43)	70 (2)	40 (1)

## Data Availability

The complete data collected in the research are available from the authors.

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
