# Peer review of "Seed Germination of Invasive Phytolacca americana and Potentially Invasive P. acinosa"

_plants, 2023, doi:10.3390/plants12051052_

Round 1
Reviewer 1 Report
The authors should bear in mind that 'seed propagation' in plants is not necessarily equal to 'sexual reproduction'. Agamospermy is an example for asexual reproduction by seeds. The whole experimental design is related to the reproduction of the two Phytolacca species by seeds, however, the authors do not provide any evidence that this is 'sexual' reproduction (i.e. involving fussion of male and female gametes and so on). In this light, the title of the article is somewhat misleading. I suggest that the title is changed to 'Seed propagation of invasive Phytolacca americana and P. acinosa'. Otherwise, the whole design of the article would be incorrect - the authors do not investigate the 'sexual' reproduction and do not provide any evidence for 'sexual processes'. The same applies to some sentences from the article where 'sexual reproduction' needs to be replaced with 'seed propagation' to be correct, e.g. lines 9 and 259 (exactly the same traits are true for agamospermous species as well, e.g. in the triploid invasive in Europe Erigeron annuus, so these traits are not specific to 'sexual' invasive plants).
On lines 155 and 183 it is stated in the legends to the figures that 'Different italic letters in legends mark statistically different curves'. However, I could not see any italic letter except for the Latin names of the species.
Author Response
Dear reviewer,
Thank you for your valuable comments. Here is our response.
Comment 1: The authors should bear in mind that 'seed propagation' in plants is not necessarily equal to 'sexual reproduction'. Agamospermy is an example for asexual reproduction by seeds. The whole experimental design is related to the reproduction of the two Phytolacca species by seeds, however, the authors do not provide any evidence that this is 'sexual' reproduction (i.e. involving fussion of male and female gametes and so on). In this light, the title of the article is somewhat misleading. I suggest that the title is changed to 'Seed propagation of invasive Phytolacca americana and P. acinosa'. Otherwise, the whole design of the article would be incorrect - the authors do not investigate the 'sexual' reproduction and do not provide any evidence for 'sexual processes'. The same applies to some sentences from the article where 'sexual reproduction' needs to be replaced with 'seed propagation' to be correct, e.g. lines 9 and 259 (exactly the same traits are true for agamospermous species as well, e.g. in the triploid invasive in Europe Erigeron annuus, so these traits are not specific to 'sexual' invasive plants).
We have replaced the phrase "sexual reproduction" in the title of the manuscript with "seed germination". Accordingly, we replaced the phrase in the text (lines 10, 32, 253).
Comment 2: On lines 155 and 183 it is stated in the legends to the figures that 'Different italic letters in legends mark statistically different curves'. However, I could not see any italic letter except for the Latin names of the species.
The small italic letters a and b are used in the legends of all eight graphs in figures 1 and 2. We changed the text in both figure captions to make it clearer.
Comment 3: The English of the manuscript should be revised
The English language in the manuscript was edited.
Reviewer 2 Report
The article is suitable for the upcoming special issue of the journal.
I have the following comments on the article.
Title
If possible, edit the article title to clearly state that only P. americana has an invasive status. As proposed, the title is misleading and makes an impression that both species are invasive. I recommend replacing this term e.g. with expression „allien“ or add species status to the title.
Materials and methods
Please, specify the exact date (part of the month) of the collection of plant material, State whether the both species were at the same stage of the fruit maturation, due to the possible different beginning of flowering and the lenght of ripening of the observed species.
Results
The legend of the Figure 4B is incomplete. Please add the formula: „The colours of the columns represent colours of fruits.“
Nowhere in the results is it stated why only the P. acinosa species was chosen to illustrate fruit ripening (fig. 4A). What was the purpose of this choice? Why is the process of ripening of P. americana species omited?
Discussion
Disscus whether the thickness of the testa could have been responsible for the different germination of fresh seeds of the studied species.
Add to the discussion whether the autumn emergence and survival of seedlings of any of monitored species was recorded.
Only the mechanical method of plant removal is recommended in the eradication protocol.
Other procedures such as e.g. chemical liquidation (e.g. dessication) are not suggested. Discuss and justify other acceptable or inappropriate methods of eradication.
Language
The English of the manuscript should be revised.
Author Response
Dear reviewer,
Thank you for your valuable comments. Here is our response.
Comment 1: If possible, edit the article title to clearly state that only P. americana has an invasive status. As proposed, the title is misleading and makes an impression that both species are invasive. I recommend replacing this term e.g. with expression "allien "or add species status to the title.
We changed the title of our manuscript, so the status of both species is more accurate.
Comment 2: Please, specify the exact date (part of the month) of the collection of plant material, State whether the both species were at the same stage of the fruit maturation, due to the possible different beginning of flowering and the lenght of ripening of the observed species.
The exact dates of the collection of plant material are included in the manuscript (lines 377-394). P. acinosa indeed starts to flower earlier than P. americana. Since the fruits on Phytolacca plants mature gradually, collecting the fruit racemes containing fruits of different ripeness of both species was still possible. We observed no difference between the species when we isolated the seeds from fruits for germination experiments.
Comment 2: The legend of the Figure 4B is incomplete. Please add the formula: "The colours of the columns represent colours of fruits. "
Done. (line 230)
Comment 3: Nowhere in the results is it stated why only the P. acinosa species was chosen to illustrate fruit ripening (fig. 4A). What was the purpose of this choice? Why is the process of ripening of P. americana species omited?
We included the series of photographs only as an illustration of fruit ripening. Very similar situations were observed in other fruit racemes of P. acinosa (except some with green fruits only) and P. americana. The comparison between the observed racemes of both species is summarised in figure 4B. We reformulated the sentence in line 213 to make it clearer.
Comment 4: Discuss whether the thickness of the testa could have been responsible for the different germination of fresh seeds of the studied species.
Both species have very similar shapes and sizes of seeds. The testa of ripe seeds of both is black and similarly thick. We observed that when cutting the seeds for the Tetrazolium tests. However, the thickness of the seed's testa was not precisely measured. We slightly changed the paragraph (lines 268-276) to stress the similarity of seeds of both Phytolacca species.
Comment 4: Add to the discussion whether the autumn emergence and survival of seedlings of any of monitored species was recorded.
Done. (lines 337-338)
Comment 4: Only the mechanical method of plant removal is recommended in the eradication protocol. Other procedures such as e.g. chemical liquidation (e.g. dessication) are not suggested. Discuss and justify other acceptable or inappropriate methods of eradication.
We added a paragraph to explain why we did not include the use of herbicides in the protocol. (lines 356-365
Comment 5: The English of the manuscript should be revised
The English language in the manuscript was edited.